# Real-time resolution of short-read assembly graph using ONT long reads

**Son Hoang Nguyen**[1]*, **Minh Duc Cao**[1], **Lachlan J. M. Coin**[1,2,3,4]*

**1** Institute for Molecular Bioscience, the University of Queensland, St Lucia, Brisbane, Australia,
**2** Department of Microbiology and Immunology, The University of Melbourne, Parkville, Australia,
**3** Department of Clinical Pathology, The University of Melbourne, Parkville, Australia, **4** Department of
Infectious Disease, Imperial College London, London, UK

\* s.hoangnguyen@imb.uq.edu.au (SHN); l.coin@imb.uq.edu.au (LC)

## Abstract

A streaming assembly pipeline utilising real-time Oxford Nanopore Technology (ONT) sequencing data is important for saving sequencing resources and reducing time-to-result. A previous approach implemented in `npScarf` provided an efficient streaming algorithm for hybrid assembly but was relatively prone to mis-assemblies compared to other graph-based methods. Here we present `npGraph`, a streaming hybrid assembly tool using the assembly graph instead of the separated pre-assembly contigs. It is able to produce more complete genome assembly by resolving the path finding problem on the assembly graph using long reads as the traversing guide. Application to synthetic and real data from bacterial isolate genomes show improved accuracy while still maintaining a low computational cost. `npGraph` also provides a graphical user interface (GUI) which provides a real-time visualisation of the progress of assembly. The tool and source code is available at https://github.com/hsnguyen/assembly.

Real-time resolution of short-read assembly graph
using ONT long reads. PLoS Comput Biol 17(1):
e1008586. https://doi.org/10.1371/journal.
pcbi.1008586

STATES

**Data Availability Statement:** The primary
benchmark simulation data can be obtained from
this link: https://cloudstor.aarnet.edu.au/plus/index.
php/s/dzRCaxLjpGpfKYW The real bacterial

## Author summary

Hybrid genome assembly algorithms combine high accuracy short reads with error-prone long reads with the goal of generating highly contiguous assemblies with low error rates. Short read sequence data is relatively inexpensive in comparison to long-read sequence data, and, moreover short-read sequence data has already been collected for many bacterial species, thus motivating development of methods wh ich are frugal with respect to acquisition of long-read sequence data. One of the attractive features of the Oxford Nanopore Technology's sequencers is that they generate sequence data in real-time, and in principle sequencing can be stopped once enough data is acquired. However, there is only one previous attempt for greedy genome scaffolding of contigs in real-time, which was prone to assembly errors. In this paper we describe a new tool—`npGraph`—which resolves the assembly graph in real-time as sequence is generated; coupled with assembly visualisation showing assembly graph resolution in real-time. We show that `npGraph` generates completed bacterial assemblies which are as accurate as state-of-the-art batch hybrid assembly pipelines, and also provides substantial computational speed-up.

sequencing data was from NCBI under project accession number PRJNA353060.

**Funding:** This study was supported by a Discovery Project with grant number DP170102626 awarded by the Australian Research council to MC and LC. The funders had no role in study design, data collection and analysis, decision to publish, or preparation of the manuscript.

**Competing interests:** I have read the journal's policy and the authors of this manuscript have the following competing interests: LC and MC have received travel funding to attend Oxford Nanopore Technologies conferences. LC has received research funding from ONT to support development of Chiron basecaller.

This is a *PLOS Computational Biology* Methods paper.

## Introduction

Sequencing technology has reached a level of maturity which allows the decoding of virtually any piece of genetic material which can be obtained. However, the time from sample to result remains a barrier to adoption of sequencing technology into time critical applications such as infectious disease diagnostics or clinical decision making. While there exists real-time sequencing technology such as Oxford Nanopore Technologies (ONT), algorithms for streaming analyses of such real-time data are still in their infancy. Effective streaming methodology will help bridge the gap between potential and practical use.

One particular strength of ONT technology is the production of ultra long reads. This is complementary to the dominant short read sequencing technology Illumina which is cheaper and has higher per base read quality but is unable to resolve the complex regions of the genome due to its read length limitation. Hybrid assembly with ONT reads provides the opportunity to use long reads to fully resolve existing microbial genome assemblies. However, in order achieve this in an effective manner it is necessary to avoid both under-sequencing (and thus not completely resolving the assembly) or over-sequencing (and thus incurring unnecessary costs and time-to -results). In addition, the release of Read Until API provides the opportunity to selectively enrich parts of the genome, as has been implemented in customised targeted sequencing applications [1]. The combination of ReadUntil with streaming hybrid assembly opens the possibility for further efficiency gains by targeting sequencing to unresolved regions of the genome.

Previously, we developed `npScarf` [2] an algorithm to scaffold a draft assembly from Illumina sequencing simultaneously with ONT sequencing. However, `npScarf` ignores the rich connectivity information in the short read assembly graph, and as a result is relatively prone to mis-assembly compared to alternative methods.

Here, we present `npGraph`, a novel algorithm to resolve the assembly graph in real-time using long read sequencing. `npGraph` uses the stream of long reads to untangle knots in the assembly graph, which is maintained in memory. Because of this, `npGraph` has better estimation of multiplicity of repeat contigs, resulting in fewer misassemblies. In addition, we develop a visualisation tool for practitioners to monitor the progress of the assembly process.

## Materials and methods

### Application overview

`npGraph` makes use of an assembly graph generated from assembling short reads using a de Bruijin graph method such as `SPAdes` [3], `Velvet` [4] and `AbySS` [5]. The assembly graph consists of a list of contigs, and possible connections among these contigs. In building the assembly graph, the de Bruijin graph assembler attempts to extend each contig as far as possible, until there is more than one possible way of extending due to the repetitive sequences beyond the information contained in short reads. Hence each contig has multiple possible connections with others, creating knots in the assembly graph. `npGraph` uses the connectivity information from long reads to untangle these knots in real-time. With sufficient data, when all the knots are removed, the assembly graph is simplified to a path which represents the complete assembly.

`npGraph` aligns long reads to the contigs in the assembly graph. When a long read is aligned to multiple contigs, `npGraph` constructs candidate paths that are supported by the

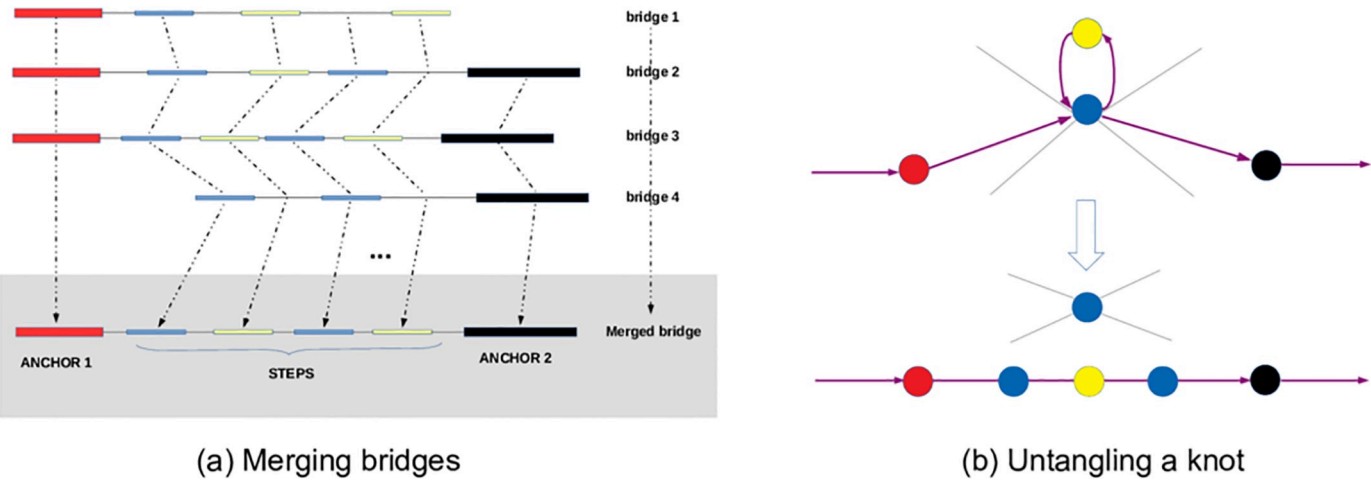

**Fig 1. Graph resolving algorithm.** (a) the bridges suggested by long reads are merged progressively with dynamic programming to find the best path connecting 2 anchors. (b) A knot (repetitive contig) is unwound following the best path (highlighted in purple) leading to the graph simplification.

read. This strategy allows `npGraph` to progressively update the likelihood of the paths going through a knot. When sufficient data are obtained, the best path is confidently identified and hence the knot is untangled. In general, the bridging algorithm used to determine the best path is a combination of progressive merging, accumulated scoring and decision making modules. It operates on each pair of unique contigs, or *anchors*, by using a *bridge* data structure maintaining 2 anchors and a list of *steps* in-between as shown in Fig 1A. A set of candidate paths are listed and the best one can be selected amongst them given enough evidence. The de Bruijin graph is subsequently simplified when the bridge is replaced by a single edge representing the best path (Fig 1B).

We also provide a Graphical User Interface (GUI) for `npGraph`. The GUI includes the dashboard for controlling the settings of the program and a window for visualization of the assembly graph in real-time (Fig 2). In this interface, the assembly graph loading stage is separated from the actual assembly process so that users can check for the graph quality first before carry out any further tasks. A proper combination of command line and GUI can provide an useful streaming pipeline that copes well with MinION output data. This is designed to support the real-time monitoring of the results from real-time sequencing [2, 6, 7] that allow the analysis to take place abreast to a nanopore sequencing run.

## Algorithm details

The work flow of `npGraph` mainly consists of 3 stages: (1) assembly graph pre-processing; (2) graph resolving and simplifying; (3) post-processing and reporting results. The first step is to load the assembly graph of Illumina contigs and analyze its components' property, including binning and multiplicity estimation. The second step works on the processed graph and the long read data that can be provided in real-time by ONT sequencer. Based on the paths induced from long reads, the assembly graph will be resolved on-the-fly. Finally, the graph is subjected to the last attempt of resolving and cleaning, as well as output the final results. The whole process can be managed by using either command-line interface or GUI. Among three phases, only the first one must be performed prior to the MinION sequencing process in a streaming setup. The algorithm works on the assembly graph of Illumina contigs, so the terms *contigs* and *nodes* if not mentioned specifically, would be used interchangeable throughout this context.

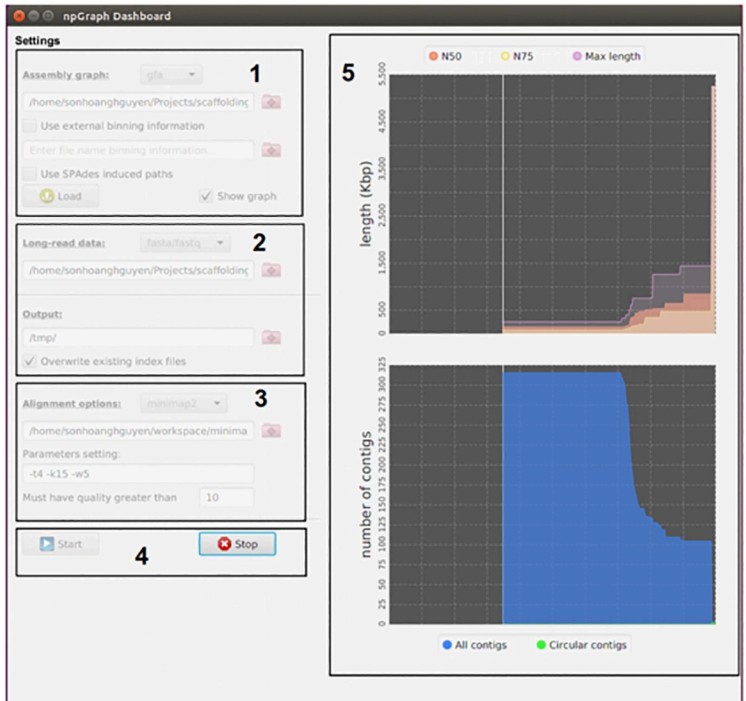
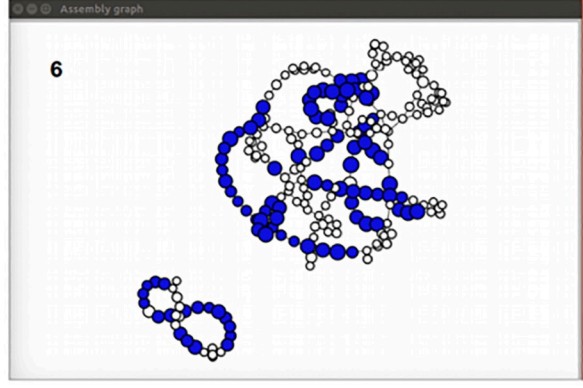
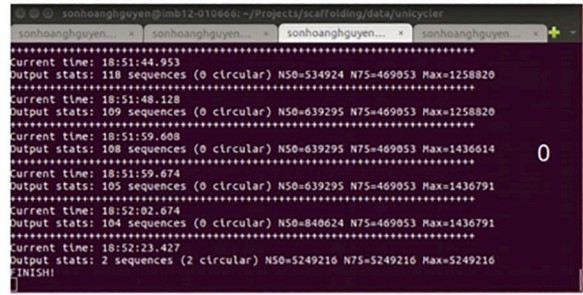

**Fig 2. `npGraph` user interface including Console (0) and GUI components (1-6).** The GUI consists of the Dashboard (**1-5**) and the Graph View (**6**). From the Dashboard there are 5 components as follow: **1** the assembly graph input field; **2** the long reads input field; **3** the aligner settings field; **4** control buttons (start/stop) to monitor the real-time scaffolding process; **5** the statistics plots for the assembly result.

**Contigs binning.** Contigs should belong to single or multiple groups, or *bins*, that would represent different assembly units, *e.g.* chromosome, plasmids, of different species if applying to a metagenomics dataset. A binning step is needed to assign membership of each contig to its corresponding group. This information can aid the bridging stage later, as member contigs of a bridge are expected to appear in the same group. However, the criteria can be relaxed between closed groups given strong connecting evidence from long read alignments.

The first step in our binning procedure is to cluster the big anchors (longer than 10Kbp and in/out degree less than 2) based on their *kmer* coverage. To achieve this, we applied DBSCAN clustering algorithm [8], which utilises a customised metric function to map contigs into a one-dimensional space. In order to define a customised metric which is simple and fast to calculate, we assumed that a single long contig itself consists of a Poisson distribution of *k-mers* count with the mean approximated by the contig's coverage. The metric is then determined by a distance function of two Poisson distributions based on Kullback-Leibler divergence (or relative entropy) between the Poisson distribution representing each contig [9].

Formally, assuming there are 2 Poisson distributions $P_1$ and $P_2$ with probability mass functions (PMF)

$$P_1(X = x, \lambda_1) = \frac{e^{-\lambda_1}\lambda_1^x}{x!}$$

and

$$P_2(X = x, \lambda_2) = \frac{e^{-\lambda_2}\lambda_2^x}{x!}$$

The Kullback-Leibler divergence from $P_2$ to $P_1$ is defined as:

$$D_{KL}(P_1||P_2) = \sum_x P_1(x) \log \frac{P_1(x)}{P_2(x)} = \mathbb{E}_{P_1}[\log \frac{P_1(x)}{P_2(x)}]$$

or in other words, it is the expectation of the logarithmic difference between the probabilities $P_1$ and $P_2$, where the expectation is taken using $P_1$. The log ratio of the PMFs is:

$$\log \frac{P_1(x)}{P_2(x)} = \log(e^{\lambda_2 - \lambda_1}(\frac{\lambda_1}{\lambda_2})^x) = x \log \frac{\lambda_1}{\lambda_2} + \lambda_2 - \lambda_1$$

Thus the divergence between $P_1$ and $P_2$ is:

$$D_{KL}(P_1||P_2) = \mathbb{E}_{P_1}[\log \frac{P_1(x)}{P_2(x)}] = \lambda_1 \log \frac{\lambda_1}{\lambda_2} + \lambda_2 - \lambda_1$$

Thus, the metric we used is a distance function defined as:

$$D(P_1, P_2) = \frac{D_{KL}(P_1||P_2) + D_{KL}(P_2||P_1)}{2} = \frac{1}{2}(\lambda_1 - \lambda_2)(\log \lambda_1 - \log \lambda_2)$$

Independent from the contigs clustering in the pre-processing step, additional evidence of nodes' uniqueness can be acquired using the long reads during the assembly process. Given enough data, the multiplicity of an ambiguous node can be determined based on the set of all bridges rooted from itself. On the other hand, external binning tools such as MetaBAT [10], maxbin [11] can be employed in `npGraph` as well.

**Multiplicity estimation.** Now bins of the main unique contigs had been identified, however, they only make up a certain proportion of the contigs set. From here, we need to assign bin membership and multiplicity for all other nodes of the graph, especially the repetitive ones. To do so, we relied on the graph's topology and the estimated read coverage of initial contigs from SPAdes. Given all contigs' coverage values as nodes' weight, we need to estimate those of edges and in return, using them to re-estimate the coverage for repetitive nodes if necessary. After this process, we will have a graph with optimized weighted components that would suggest their multiplicities more exactly. Basically the computation is described as in following steps:

0. Initialize every node weight as its corresponding contig coverage, all edges' weight as zeros.

1. Calculate distributed weights for edges by quadratic unconstrained optimization of the least-square function:

$$\frac{1}{2}\sum_i l_i((\sum e_i^+ - c_i)^2 + (\sum e_i^- - c_i)^2)$$

where $l_i$ and $c_i$ is the length and weight of a node $i$ in the graph;
$\sum e_i^+$ and $\sum e_i^-$ indicates sum of weights for incoming and outgoing edges from node $i$ respectively. They are expected to be as close to $c_i$ as possible thus the length-weighted least-square should be minimized.
The above function can be rewritten as:

$$f(x) = \frac{1}{2}x^T Q x + b^T x + r$$

and then being minimized by using gradient method.

2. Re-estimate weights of repetitive nodes based on their neighboring edges' measures and repeat previous optimization step. The weights are calculated iteratively until no further significant updates are made or a threshold of loop count is reached.

At this point, we can induce the copy numbers of nodes in the final assembly. For each node, this could be done by investigating its adjacent edges' multiplicity to estimate how many times it should be visited and from which bin(s). Multiplicities of insignificant nodes (of sequences with length less than 1, 000 bp) are less confident due to greater randomness in sequencing coverage. For that reason, in npGraph, we did not rely on them for graph transformation but as supporting information for path finding.

**Building bridges in real-time.**   Bridge is the data structure designed to identify the possible connections between two anchored nodes (of unique contigs) in the assembly graph. A bridge must start from a unique contig, or *anchor* node, and end at another if completed. Located in-between are nodes known as *steps* and distances between them are called *spans* of the bridge. Stepping nodes are normally repetitive contigs and indicative for a path finding operation later on. In a complicated assembly graph, the more details the bridge, *a.k.a.* more steps in-between, the faster and more accurate the linking path it would resolve. A bridge's function is complete when it successfully return the ultimate linking path between 2 anchors.

The real-time bridging method considers the dynamic aspect of multiplicity measures for each node, meaning that a *n*-times repetitive node might become a unique node at certain time point when its $(n - 1)$ occurrences have been already identified in other distinct unique paths. Furthermore, the streaming fashion of this method allows the bridge constructions (updating steps and spans) to be carried out progressively so that assembly decisions can be made immediately after having sufficient supporting data. A bridge in npGraph has several completion levels. When created, it must be rooted from an *anchor node* which represents a unique contig (level 1). A bridge is known as fully complete (level 4) if and only if there is a unique path connecting its two anchor nodes from two ends.

At early stages (level 1 or 2), a bridge is constructed progressively by alignments from long reads that spanning its corresponding anchor(s). In an example from Fig 1A, bridges from a certain anchor (highlighted in red) are created by extracting appropriate alignments from incoming long reads to the contigs. Each of the steps therefore is assigned a weighing score based on its alignment quality. Due to the error rate of long reads, there should be deviations in terms of steps found and spans measured between these bridges, even though they represent the same connection. A continuous merging phase, as shown in the figure, takes advantage of a pairwise Needleman-Wunsch dynamic programming to generate a consensus list based on weight and position of each of every stepping nodes. The spans are calibrated accordingly by averaging out the distances. On the other hand, the score of the merged steps are accumulated over time as well. Whenever a consensus bridge is anchored by 2 unique contigs at both ends and hosting a list of steps with sufficient coverage, it is ready for a path finding in the next step.

**Path finding algorithm.**   Given a proposed bridge with 2 anchors $B = \{\overrightarrow{v_0} \ldots, \overrightarrow{v_n}\}$ identified from previous step, a searching algorithm is implemented to find the final bridging solution, *i.e.* an ungapped path among all possiblities. To do so, each of the candidates is given a score of alignment-based likelihood which are updated immediately as long as there is an appropriate long read being generated by the sequencer. As more nanopore data arrives, the score divergence increases and only the highest one would be selected for the next round.

Overall, the algorithm employs a binary search strategy as shown in Algorithm 1. Starting with the original bridge $\{\overrightarrow{v_0} \ldots, \overrightarrow{v_n}\}$, we recursively break the target bridge in half at one of its *step* $\overrightarrow{v_k}$ in-between then find the sub-paths of $\{\overrightarrow{v_0} \ldots, \overrightarrow{v_k}\}$ and $\{\overrightarrow{v_k} \ldots, \overrightarrow{v_n}\}$. Algorithm 1 will

call Algorithm 2 when the base condition is met, meaning the target bridge is unbreakable (no more in-between *step* available).

**Algorithm 1**: Recursive binary bridging to connect 2 anchor nodes.

**Data**: Assembly graph $G\{V, E\}$

**Input**: Brigde $B : \{\vec{v_0}, \ldots \vec{v_k}, \ldots, \vec{v_n}\}$ with $\vec{v_0}$ and $\vec{v_n}$ are two anchors, $\{\vec{v_k}\}, k = 1 \ldots (n-1)$ are steps in-between

**Output**: Set of candidate paths connecting $\vec{v_0}$ to $\vec{v_n}$ that maximize the likelihood of the step list.

1 **Function** BinaryBridging($B$):

 /* search for the contig with maximum score from the step list (two ends excluded) */

2 $m := \mathrm{argmax}_k(\vec{v_k}.score())$

 /* if there is no step in-between, run Algorithm 2 and return the result */

3 **if** $M.size() \equiv 2$ **then**

4 **return** DFS($B.start()$, $B.end().B.distance()$)

 /* break the original bridge $B$ into 2 bridges by $v_m$: *BL* and *BR* */

5 $BL := \{B.start(), \ldots, \vec{v_m}\}$

6 $BR := \{\vec{v_m}, \ldots, B.end()\}$

 /* recursively invoke the procedure and join the results together */

7 **return** BinaryBridging($BL$) $\bowtie$ BinaryBridging($BR$)

In Algorithm 2, we implement a modified stack-based Depth-First Search (DFS) using Dijkstra's shortest path finding algorithm [12] to reduce the search space. In which, function

$$\mathtt{shortestTree}(\overrightarrow{vertex}, distance) : (V, Z) \to V^n$$

from line 3 builds a shortest tree rooted from $\vec{v}$, following its direction until a distance of approximately $d$ (with a tolerance for the nanopore error rate) is reached, similar to the Dijkstra algorithm. This tree is used on line 4 and in function *includedIn*() on line 19 to filter out any node or edge with ending nodes that do not belong to the tree.

**Algorithm 2**: Pseudo-code for finding paths connecting 2 nodes given their estimated distance.

**Data**: Assembly graph $G\{V, E\}$

**Input**: Pair of bidirected nodes $\vec{v_1}, \vec{v_2}$ and estimated distance $d$ between them

**Output**: Set of candidate paths connecting $\vec{v_1}$ to $\vec{v_2}$ with reasonable distances compared to $d$

1 **Function** DFS($\vec{v_1}, \vec{v_2}$, $d$):

2 $P :=$ new List()

3 $M := \mathtt{shortestTree}(\vec{v_2}, d)$ // build shortest tree from $\vec{v_2}$ with range $d$

4 **if** $M.contain(\vec{v_1})$ **then**

5 $S :=$ new *Stack*() // stack of sets of edges to traverse

6 $edgesSet := getEdges(\vec{v_1})$ // get all bidirected edges going from $\vec{v_1}$

7 $S.push(edgesSet)$

8 $p :=$ new *Path*($\vec{v_1}$) // init a path that has $\vec{v_1}$ as root

9 **while** *true* **do**

10 $edgesSet := S.peek()$

11 **if** $edgesSet.isEmpty()$ **then**

12 **if** $p.size() \leq 1$ **then**

13 **break** // stop the loop when there is no more edge to discover

14 $S.pop()$

15 $d += p.peekNode.length() + p.popEdge().length()$

16 **else**

17 $curEdge := edgesSet.remove()$

18 $\vec{v} := curEdge.getOpposite(p.peekNode())$

19 $S.push(getEdges(\vec{v}).includedIn(M))$

```
20              p.add(curEdge)
21              if reach v⃗₂ with reasonable d then
22                P.add(p)
23              d− = v⃗.length() + curEdge.length()
24    return P
```

Basically, the algorithm keeps track of a stack that contains sets of candidate edges to discover. During the traversal, a variable $d$ is updated as an estimation for the distance to the target. A hit is reported if the target node is reached with a reasonable distance *i.e.* close to zero, within a given tolerance (line 21). Another threshold for the maximum traversing depth is set and heuristic cut-off can be used to preven exceptional combinatorial explosion when traversing extremely complex graph components.

It worths noting that from the pseudo code in Algorithm 2, despite of identical name, procedure *length*() used for nodes are different than for edges. The former would return the contig length while the latter measure the overlap or distance between a pair of contigs thus can be negative or positive respectively. From De Bruijin graph for instance, all edges have same the length value of $-k$. In npGraph's algorithm, an edge's length can vary as we allow additional connection types on top of the original SPAdes assembly graph input.

In many cases, due to dead-ends, there not always exist a path in the assembly graph connecting two anchors as suggested by the alignments. In this case, if enough long reads coverage (20X) are met, a consensus module is invoked and the resulting sequence is contained in a *pseudo* edge.

**Graph simplification in real-time.**   npGraph resolves the graph by reducing its complexity perpetually using the long reads that can be streamed in real-time. Whenever a bridge is finished (with a unique linking path), the assembly graph is *transformed* or *reduced* by replacing its unique path with a composite edge and removing any unique edges (edges coming from unique nodes) along the path. The assembly graph would have at least one edge less than the original after the reduction. The nodes located on the reduced path, other than 2 ends, also have their multiplicities subtracted by one and the bridge is marked as finally resolved without any further modifications.

Fig 3 presents an example of the results before and after graph resolving process in the GUI. The result graph, after cleaning, would only report the significant connected components that represents the final contigs. Smaller fragments, even unfinished but with high remaining coverage, are also presented as potential candidates for further downstream analysis. Further annotation utility can be implemented in the future better monitoring the features of interests as in npScarf.

**Result extraction and output.**   npGraph reports assembly result in real-time by decomposing the assembly graph into a set of longest straight paths (LSP), each of the LSP will spell a contig in the assembly report. The final assembly output contains files in both FASTA and GFAv1 format (https://github.com/GFA-spec/GFA-spec). While the former only retains the actual genome sequences from the final decomposed graph, the latter output file can store almost every properties of the ultimate graph such as nodes, links and potential paths between them.

A path $p = \{v_0, e_1, v_1, \ldots, v_{k-1}, e_k, v_k\}$ of size $k$ is considered as straight if and only if each of every edges along the path $e_i, \forall i = 1, \ldots, k$ is the only option to traverse from either $v_{i-1}$ or $v_i$, giving the transition rule. To decompose the graph, the tool simply mask out all incoming/outgoing edges rooted from any node with in/out degree greater than 1 as demonstrated in Fig 4. These edges are defined as branching edges which stop straight paths from further extending.

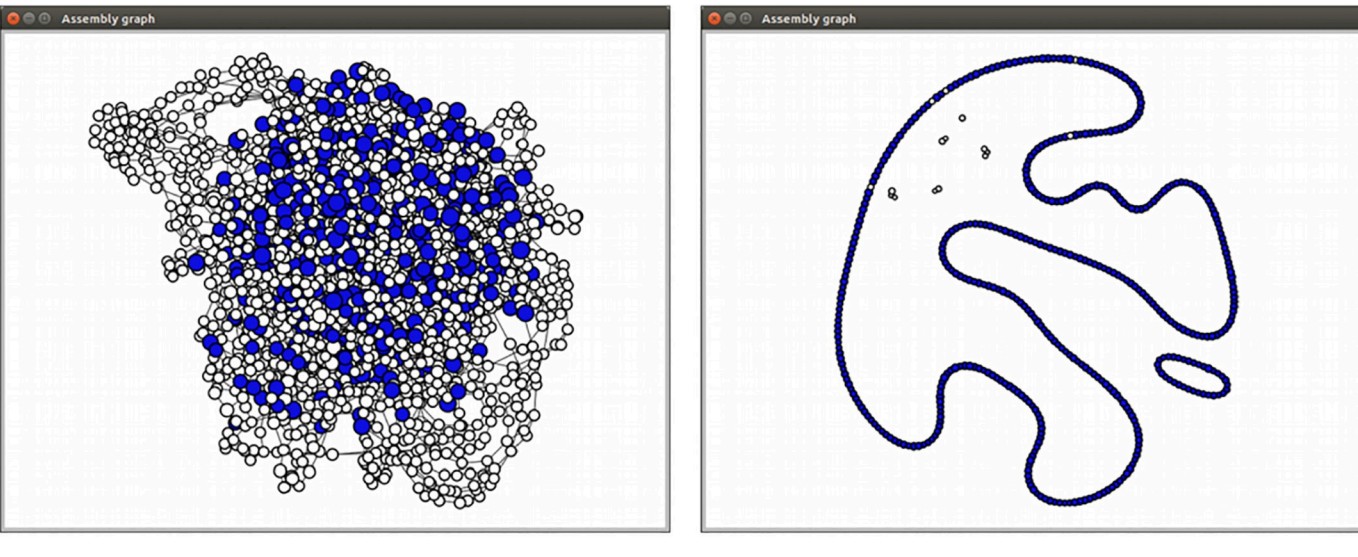

(a) Initial graph   (b) Resolved graph

**Fig 3. Assembly graph of *Shigella dysenteriae* Sd197 synthetic data being resolved by `npGraph` and displayed on the GUI Graph View.** The `SPAdes` assembly graph contains 2186 nodes and 3061 edges, after the assembly shows 2 circular paths representing the chromosome and one plasmid.

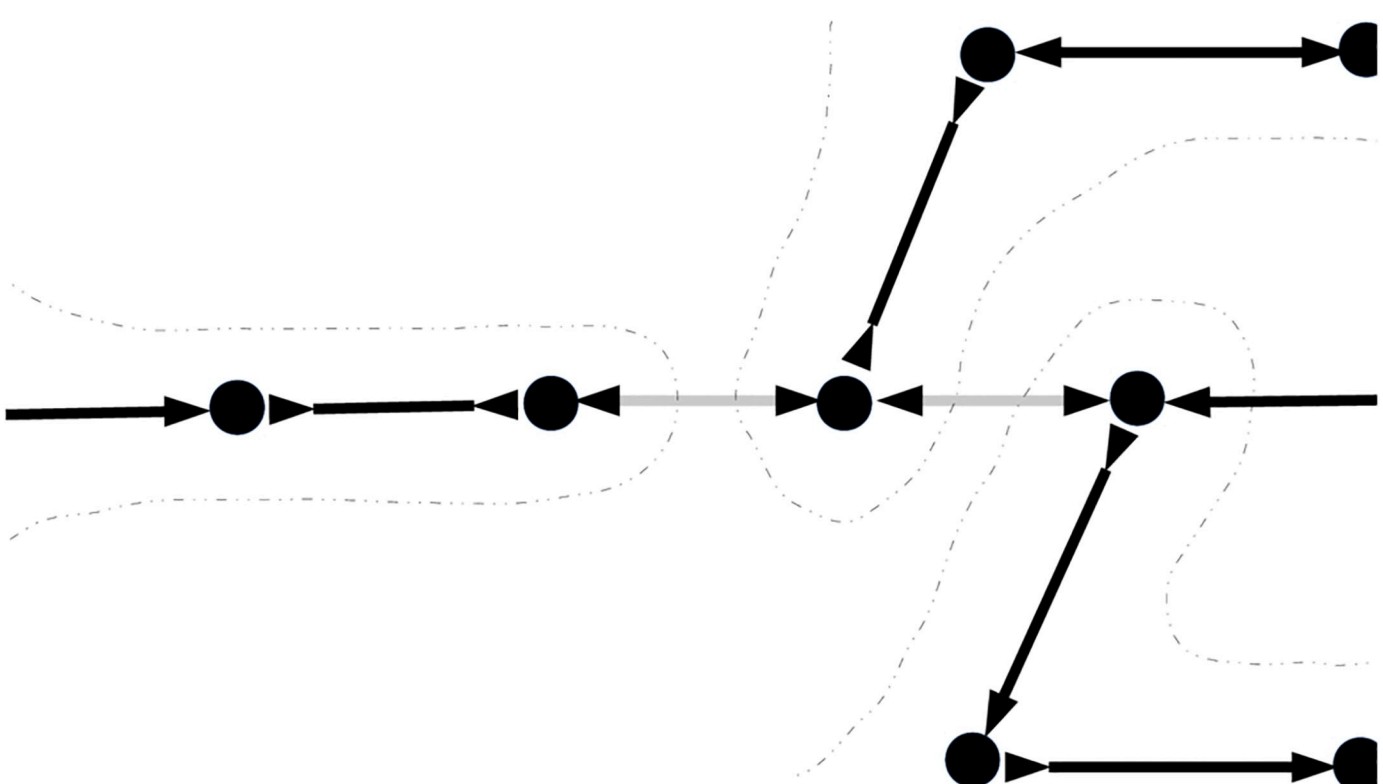

**Fig 4. Example of graph decomposition into longest straight paths.** Branching edges are masked out (faded) leaving only straight paths (bold) to report. There would be 3 contigs extracted by traversing along the straight paths here.

The decomposed graph is only used to report the contigs that can be extracted from an assembly graph at certain time point. For that reason, the branching edges are only masked but not removed from the original graph as they would be used for further bridging.

Other than that, if GUI mode is enabled, basic assembly statistics such as N50, N75, maximal contigs length, number of contigs can be visually reported to the users in real-time beside the Dashboard. The progressive simplification of the assembly graph can also be observed at the same time in the Graph view.

## Results

### Evaluation using synthetic data

To evaluate the performance of the method, `npGraph` was tested along with `SPAdes`, `SPAdes` hybrid from version 3.13.1, [13], `npScarf` (japsa 1.7-02), `LRScaf` version 1.1.9 [14] and `Unicycler` version 0.4.6 using `Unicycler`'s synthetic data set [15]. The data set is a simulation of Illumina and MinION raw data, generated *in silico* based on available microbial references. We ran hybrid assembly methods using the entire nanopore data and the reciprocal results were evaluated by QUAST 5.0.2 [16].

Table 1 shows comparative results running different methods on 5 synthetic data sets, simulated from complete genomes of *Mycobacterium tuberculosis* H37Rv, *Klebsiella pneumoniae* 30660/NJST258_1, *Saccharomyces cerevisiae* S288c, *Shigella sonnei* 53G and *Shigella dysenteriae* Sd197. The nanopore read depth for each data set is approximately 65-fold coverage even though much less data were needed to generate these final results by the aforementioned hybrid methods.

To align the long reads to the assembly graph components, both `BWA-MEM` [17] or `minimap2` [18] were used in conjunction with `npGraph`. These two methods were chosen due to their proven efficiency and compatibility with streaming data. While `BWA-MEM` is a well-known classic aligner that can be adapted to work with third generation sequencing data, `minimap2` has been specially designed for this data type. We observed a slightly higher error rate (comprising the sum of mismatches and indels per 100kb) using `BWA-MEM` in comparison to `minimap2` for all simulations in Table 1. This is due to the fact that bridging paths induced using `BWA-MEM` were slightly less accurate due to more noise from the smaller *steps* in-between (Fig 1A). However, under almost circumstances, using either aligner resulted in final assemblies with comparable qualities. In terms of running time and resources required, `minimap2` always proved to be a better option, requiring markedly less CPU time than `BWA-MEM`. Utilising `minimap2`, `npGraph` is now the fastest hybrid assembler available.

Amongst all assemblers, `Unicycler` applies an algorithm based on semi-global (or glocal) alignments [19] with the consensus long reads generated with the `SeqAn` library. With all of the data sets tested, `Unicycler` required the most computational resources, but it also returned fewer mis-assemblies than the other approaches with a comparable rate of error (indels and mismatches) to `npGraph`. `hybridSPAdes` reported decent results with high fidelity at base level. As the trade-off, there were fewer connections satisfying its quality threshold, resulting in the fragmented assemblies with lower N50 compared to the other hybrid assemblers. This behaviour was clearly reflected in the last, also the most challenging task of assembly *S. dysenteriae*.

Of the two streaming algorithms, `npScarf` utilizes a fast but greedy scaffolding approach that can lead to mis-assemblies and errors. For bacterial genomes with modest complexity these are minimal (e.g. *K. pneumoniae*), but for those with severe repetitive elements, extra calibrations are needed to prevent the mis-assembly due to ambiguous alignments. On the other hand, `npGraph` significantly reduced the errors compared to `npScarf`, sometimes even

**Table 1. Comparison of assemblies produced in batch-mode using `npGraph` and other hybrid assembly methods on synthetic data.**

| Method | Assembly size (Mbp) | #Contigs | N50 (Kbp) | Mis-assemblies | Error (per 100 Kbp) | Run times (CPU hrs) |
|---|---|---|---|---|---|---|
| *M. tuberculosis* H37Rv | 4,411,532 bp | | | | | |
| SPAdes | 4.376 | 66 | 150.7 | 0 | 0.23 | 1.42 |
| SPAdes hybrid | 4.411 | 1 | 4410.5 | 0 | 0.86 | 1.61 |
| Unicycler | 4.412 | 1 | 4411.5 | 0 | 2.56 | 5.52 |
| npScarf | 4.432 | 4 | 4402.2 | 7 | 6.61 | 1.42 + 0.7 |
| npGraph (bwa) | 4.411 | 1 | 4411.4 | 0 | 2.63 | 1.42 + 0.64 |
| npGraph (minimap2) | 4.412 | 1 | 4411.5 | 0 | 0.68 | 1.42 + 0.02 |
| *K. pneumoniae* 30660 | 5,540,936 bp | | | | | |
| SPAdes | 5.469 | 64 | 270.2 | 0 | 0.07 | 1.36 |
| SPAdes hybrid | 5.543 | 8 | 4229.1 | 2 | 5.04 | 1.63 |
| Unicycler | 5.538 | 9 | 5263.2 | 0 | 1.85 | 4.34 |
| npScarf | 5.566 | 7 | 5259.1 | 4 | 35.6 | 1.36 + 0.95 |
| npGraph (bwa) | 5.535 | 5 | 5263.2 | 1 | 4.16 | 1.36 + 0.92 |
| npGraph (minimap2) | 5.541 | 6 | 5263.2 | 0 | 0.85 | 1.36 + 0.04 |
| *S. cerevisiae* S288c | 12,157,105 bp | | | | | |
| SPAdes | 11.675 | 194 | 260.5 | 0 | 1.57 | 3.61 |
| SPAdes hybrid | 11.910 | 45 | 770.5 | 5 | 34.52 | 4.15 |
| Unicycler | 11.837 | 29 | 909.1 | 0 | 22.83 | 16.34 |
| npScarf | 11.990 | 22 | 796.8 | 53 | 85.5 | 3.61 + 4.35 |
| npGraph (bwa) | 12.000 | 151 | 913.1 | 3 | 38.68 | 3.61 + 4.12 |
| npGraph (minimap2) | 12.008 | 148 | 913.1 | 5 | 25.32 | 3.61 + 0.13 |
| *S. sonnei* 53G | 5,220,473 bp | | | | | |
| SPAdes | 4.796 | 392 | 27.7 | 0 | 0.44 | 1.1 |
| SPAdes hybrid | 5.218 | 8 | 2195.5 | 2 | 41.98 | 1.36 |
| Unicycler | 5.221 | 5 | 4988.5 | 0 | 7.91 | 9.64 |
| npScarf | 6.426 | 20 | 1293.8 | 84 | 366.04 | 1.1 + 0.52 |
| npGraph (bwa) | 5.293 | 97 | 4988.5 | 3 | 14.87 | 1.1 + 0.57 |
| npGraph (minimap2) | 5.293 | 97 | 4988.5 | 3 | 8.31 | 1.1 + 0.08 |
| *S. dysenteriae* Sd197 | 4,560,911 bp | | | | | |
| SPAdes | 4.096 | 534 | 14.4 | 1 | 0.68 | 1.19 |
| SPAdes hybrid | 4.486 | 23 | 821.2 | 96 | 10.99 | 1.89 |
| Unicycler | 4.561 | 3 | 4369.2 | 0 | 12.96 | 8.46 |
| npScarf | - | - | - | - | - | 1.19 + - |
| npGraph (bwa) | 4.553 | 3 | 4369.1 | 6 | 91.03 | 1.19 + 0.76 |
| npGraph (minimap2) | 4.548 | 3 | 4364.1 | 8 | 83.68 | 1.19 + 0.14 |

proved to be the best option *e.g.* for *M. tuberculosis* and *K. pneumoniae*. For the yeast *S. cerevisiae* data set, the `npGraph` assembly best covered the reference genome but the number of mis-assemblies was up to 5. The unfavourable figures, namely mis-assemblies and error, were still high in case of *S. dysenteriae*, due to the complicated and extremely fragmented graph components containing a large number of small-scaled contigs that were difficult to map with nanopore data. The progressive path finding module tried to induce the most likely solution from a stream of coarse-grained alignments, without fully succeeding.

`LRScaf` has been designed as a computationally efficient hybrid assembly tool which is scalable to large genomes. We ran `LRScaf` using default parameters on the extended benchmarking set presented in S1 Table. We observed that `LRScaf` assemblies had fewer

misassemblies than npScarf, and also exhibited a low mismatch and indel rate, but produced more fragmented assemblies than either npGraph or Unicycler.

## Hybrid assembly for real data sets

A number of sequencing data sets from *in vitro* bacterial samples [20] were used to further explore differences in performance between npGraph and Unicycler. The data included both Illumina paired-end reads and MinION sequencing based-call data for each sample. Due to the unavailability of reference genomes, there were fewer statistics reported by QUAST for the comparison of the results. Instead, we investigated the number of circular sequences and PlasmidFinder 1.3 [21] mappings to obtain an evaluation on the accuracy and completeness of the assemblies (Table 2) on three data sets of bacterial species *Citrobacter freundii*, *Enterobacter cloacae* and *Klebsiella oxytoca*.

There was high similarity between final contigs generated by two assemblers on all of these datasets. For the *Citrobacter freundii* dataset, they share the same number of circular sequences, including the chromosomal and other six replicons contigs in the, with only 48 nucleotides difference in the length of the main chromosome. Five out of six identical replicons could be confirmed as plasmids based on the occurence of origin of replication sequences from the PlasmidFinder database. In detail, two megaplasmids (longer than 100Kbp) were classified as IncFIB while the other two mid-size replicons, 85.6Kbp and 43.6Kbp, were incL and repA respectively, leaving the shortest one with 2Kbp of length as ColRNAI plasmid. The remaining circular sequence without any hits to the database was 3.2Kbp long suggesting that it could be phage or a cryptic plasmid. Both assemblers had 14.5Kbp of unfinished sequences split amongst 3 linear contigs from Unicycler and 2 for npGraph.

The assembly task for *Enterobacter cloacae* was more challenging and the chromosomal DNA remained fragmented in two contigs for both methods (of length 3.324Mbp and

**Table 2. Assembly of real data sets using Unicycler and npGraph with the optimized SPAdes output.** Assembly of real data sets using Unicycler and npGraph with the optimized SPAdes output. Circular contigs are highlighted in **bold**, fragmented assemblies are presented as X|Y where X is the total length and Y is the number of supposed contigs making up X.

| | Unicycler | npGraph | Replicons (based on PlasmidFinder 1.3) |
|---|---|---|---|
| *Citrobacter freundii* CAV1741 | **5,029,534** | **5,029,486** | Chromosome |
| | **109688** | **109688** | IncFIB(pHCM2)_1_pHCM2_AL513384 |
| | **100,873** | **100,873** | IncFIB(pB171)_1_pB171_AB024946 |
| | **85,575** | **85,575** | IncL/M(pMU407)_1_pMU407_U27345 |
| | **43,621** | **43,621** | repA_1_pKPC-2_CP013325 |
| | **3,223** | **3,223** | - |
| | **1,916** | **1,916** | ColRNAI_1__DQ298019 |
| | 14,464\|3 | 14,456\|2 | - |
| *Enterobacter cloacae* CAV1411 | 4,806,666\|2 | 4,858,438\|2 | Chromosome |
| | **90,451** | 90,693\|2 | IncR_1__DQ449578 |
| | **33,610** | **33,610** | repA_1_pKPC-2_CP013325 |
| | 13,129\|2 | 14,542\|4 | - |
| *Klebsiella oxytoca* CAV1015 | 6,153,947\|5 | **6,155,762** | Chromosome |
| | **113,105** | **113,105** | IncFII(SARC14)_1_SARC14_JQ418540; IncFII(S)_1__CP000858 |
| | **111,395** | **111,395** | - |
| | **108,418** | 109,209\|13 | IncFIB(K)_1_Kpn3_JN233704 |
| | **76,183** | **76,186** | IncL/M(pMU407)_1_pMU407_U27345 |
| | **11,638** | 11,892\|2 | - |

1.534Mbp for `npGraph` compared to 2.829Mbp and 1.978Mbp for `Unicycler`). Both methods detected two plasmids (IncR and repA), and `Unicycler` returned comlpete circular sequences for both plasmids, while `npGraph` returned circular sequence for one plasmid, while the other was fragmented into two contigs. Similar to the assembly of *Citrobacter freundii*, there was around 14Kbp of data which was unable to be finished by the assemblers (split into 2 and 4 contigs for `Unicycler` and `npGraph` respectively).

Finally, the assembly for *Klebsiella oxytoca* saw fragmented chromosome using `Unicycler` (with 5 contigs) which was a fully complete single contig for `npGraph` with 6.156Mbp of size. The two assemblers shared 3 common circular sequences of which two were confirmed plasmids. The first identical sequence represented a megaplasmid ($\simeq$ 113Kbp) with two copies of IncFII origin of replication DNA being identified. The other 76Kbp plasmid circularised by both was IncL/M with of length. The third circular contig of length 111Kbp returned no hits to the plasmid database, suggesting the importance of *de novo* replicon assembly in combination with further interrogation. `Unicycler` detected another megaplasmid of size 108.4Kbp which was fractured by `npGraph`. A fragmented contig was also observed in `npGraph` for the final contig of length 11.6Kbp where it failed to combine two smaller sequences into one.

In addition to what is presented in Table 2, dot plots for the pair-wise alignments between the assembly contigs were generated and can be found in S1 Fig. This identified a structural difference between `npGraph` and `Unicycler` assembly for the *E. cloacae* CAV1411 genome assembly. This was caused by the inconsistency of a fragment's direction on the final output contigs. Comparison to a reference genome from the same bacteria strain (GenBank ID: CP011581.1 [22]), demonstrated that contigs generated by `npGraph` produced consistent alignment, but not those generated by `Unicycler` (S2 Fig). However, we cannot at this stage rule out genuine structural variation between the two samples.

## Assembly performance on streaming data

In order to investigate the rate at which the two streaming hybrid assembly algorithms completed bacterial assemblies, we plot the N50 as a function of long-read coverage on the 4 datasets described in the previous section (Fig 5). This revealed that `npGraph` and `npScarf` both converge to the same ultimate completeness but at different rates. `npScarf` converged more quickly than `npGraph`, due to the fact that it is able to build bridges with only 1 spanning long-read, whereas `npGraph` requires 3 reads. Unlike `npScarf` where the connections could be undone and rectified later if needed, a bridge in `npGraph` will remain unchanged once created. The plot for *E. coli* data clarifies this behaviour when a fluctuation can be observed in `npScarf` assembly at $\simeq$ 3-folds data coverage. On the other hand, the N50 length of `npGraph` is always a monotonic increasing function. The sharp *jumping* patterns suggested that the linking information from long-read data had been stored and exploited at certain time point decided by the algorithm. In addition, at the end of the streaming when the sequencing is finished, `npGraph` will try for the last time to connect bridges with less than 3 supporting reads which are otherwise not part of conflicting bridges. S1 Data provides the results in detail and were used to generate Fig 5.

## Discussion

Streaming assembly methods have been proven to be useful in saving time and resources compared to conventional batch algorithms with examples including *e.g.* `Faucet` [23] and `npScarf` [2]. The first method allows the assembly graph to be constructed incrementally as long as reads are retrieved and processed. This practice is helpful dealing with huge short-read data set because it can significantly reduce the local storage for the reads, as well as save time

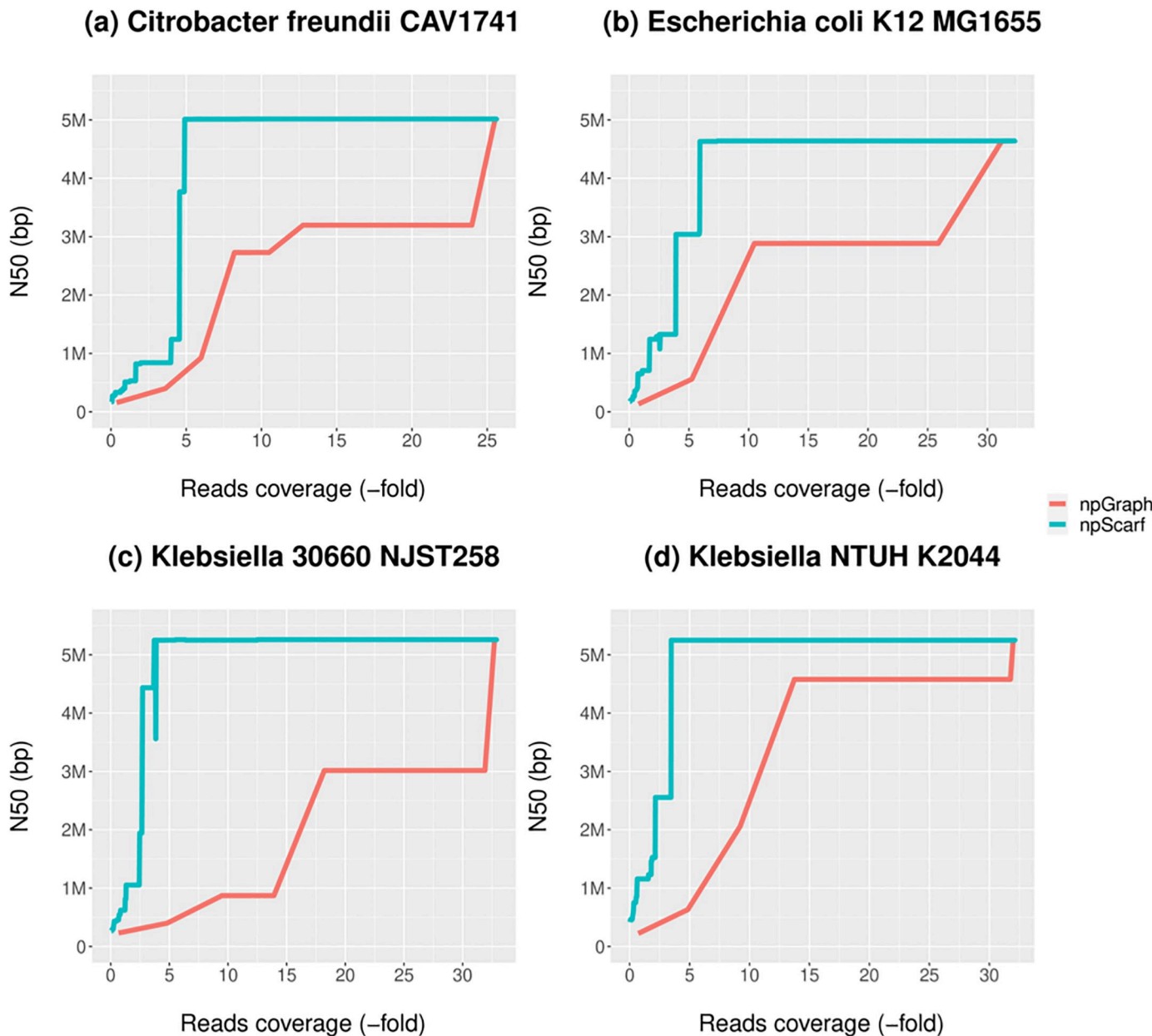

**Fig 5. N50 statistics of real-time assembly by `npScarf` and `npGraph` for 4 bacterial samples.**

for a De Bruijn graph (DBG) construction while waiting for the data being retrieved. `npScarf`, on the other hand, is a hybrid assembler working on a pre-assembly set of short-read assembly contigs. It functions by scaffolding the contigs using real-time nanopore sequencing. The completion of genome assembly in parallel with the sequencing run provides explicit benefits in term of resource control and turn-around time for analysis [2].

Hybrid approaches are still common practice in genome assembly and data analyses while Illumina sequencing retains cost and accuracy benefits over long-read sequencing. On the other hand, the third-generation sequencing methods such as Pacbio or Oxford Nanopore Technology are well-known for the ability to produce much longer reads that can further

complete the Illumina assembly. As a consequence, it is rational to combine two sources of data together in a hybrid method that can offer accurate and complete genomes at the same time. `npScarf`, following that philosophy, had been developed and deployed on real microbial genomes.

However, due to the greedy bridging approach of the contig-based streaming algorithm, `npScarf`'s results can suffer from mis-assemblies [15, 24]. A default setting was optimized for microbial genomes input but cannot fit for all data from various experiments in practice. Also, the gap filling step has to rely on the lower accuracy nanopore reads thus the accuracy of the final assembly is also affected. To tackle the quality issue while maintaining the streaming feature of the approach, a bridging method by assembly graph traversing has been proposed in this manuscript. Our approach uses as its starting point a compact DBG assembly graph, followed by graph-traveseral, repeat resolution and identification of the longest possible unbranched paths that would represents contigs for the final assembly.

Hybrid assembler using nanopore data to resolve the graph has been implemented in `hybridSPAdes` [13] as well as `Unicycler` [15]. The available tools employ batch-mode algorithms on the whole long-read data set to generate the final genome assembly. The `SPAdes` hybrid assembly module, from its first step, exhaustively looks for the most likely paths (with minimum edit distance) on the graph for each of the long read given but only ones supported by at least two reads are retained. In the next step, these paths will be subjected to a decision-rule algorithm, namely `exSPAnder` [25], for repeat resolution by step-by-step expansion, before output the final assembly. On the other hand, `Unicycler`'s hybrid assembler will initially generate a consensus long read for each of the bridge from the batch data. The higher quality consensus reads are used to align with the assembly graph to find the best paths bridging pairs of anchored contigs. While this method employs the completeness of the data set from the very beginning for a consensus step, the former only iterates over the batch of possible paths and relies on a scoring system for the final decision of graph traversal. Hence, in theory it can be adapted to a real-time pipeline.

The challenge in adapting graph-based approaches into streaming algorithm comes mainly from building a progressive implementation for path-finding and graph reducing module. To achieve this, we apply a modified DFS (depth-first search) mechanism and a dynamic voting algorithm into an on-the-fly graph resolver.

By testing with synthetic and real data, we have shown that `npGraph` can generate assemblies of comparative quality compared to other powerful batch-mode hybrid assemblers, such as `hybridSPAdes` or `Unicycler`, while also providing the ability to build and visualise the assembly in real-time.

## Conclusion

Due to the limits of current sequencing technology, application of hybrid methods should remain a common practice in whole genome assembly for the near future. On the other hand, the ONT platforms are evolving quickly with significant improvement in terms of data accuracy and yield and cost. Beside, the real-time property of this technology has not been sufficiently exploited to match its potential benefits. `npScarf` had been introduced initially to address these issues, however, the accuracy of the assembly output was affected by its greedy alignment-based scaffolding approach. Here we present `npGraph`, a streaming hybrid assembly method working on the DBG assembly graph that is able to finish short-read assembly in real-time while minimizing the errors and mis-assemblies drastically.

Compared to `npScarf`, `npGraph` algorithm employs more rigorous approach based on graph traversal. This might reduce the assembly errors because the bridging method is more

accurate so that the reporting results are more reliable. The performance of `npGraph` is comparable to `Unicycler` while consuming much less computational resources so that it can work on streaming mode. Also, the integrated GUI allows users to visualize its animated output in a more efficient way.

On the other hand, similar to `Unicycler`, `npGraph` relies on the initial assembly graph to generate the final assembly. The algorithm operates on the assumption of a high quality assembly from a well-supplied source of short-read data for a decent assembly graph to begin with. It then consumes a just-enough amount of data from a streaming input of nanopore reads to resolve the graph. Finally, extra pre-processing and comprehensive binning on the initial graph could further improve the performance of the streaming assembler.

## Supporting information

**S1 Fig. Dotplot generated by MUMmer for assembly results of `Unicycler` versus `npGraph`.** Structural agreements between two methods were found in (a) *C.freundii* and (b) *K.oxytoca* assembly contigs. On the other hand, for (c) *E.cloacae* sample, there was a disagreement detected between 2 largest contigs given by two assembly algorithms.
(TIF)

**S2 Fig. Alignments of an *Enterobacter cloacae* reference genome to assembly sequences generated by (a) `npGraph` and (b) `Unicycler`.** The former suggests a structural variant, the latter is virtually an 1-to-1 mapping.
(TIF)

**S1 Table. Benchmarking different methods using `LRScaf`, `npScarf`, `npGraph`, `hybridSPAdes` and `Unicycler` hybrid assembler with the synthetic data set.**
(XLSX)

**S1 Data. Spreadsheet contains data used to generate Fig 5.**
(XLSX)

## Acknowledgments

This research was supported by the use of the NeCTAR Research Cloud, by QCIF and by the University of Queensland's Research Computing Centre (RCC).

## Author Contributions

**Conceptualization:** Son Hoang Nguyen, Minh Duc Cao, Lachlan J. M. Coin.

**Data curation:** Son Hoang Nguyen.

**Formal analysis:** Son Hoang Nguyen, Lachlan J. M. Coin.

**Funding acquisition:** Minh Duc Cao.

**Investigation:** Son Hoang Nguyen, Minh Duc Cao, Lachlan J. M. Coin.

**Methodology:** Son Hoang Nguyen, Minh Duc Cao.

**Project administration:** Lachlan J. M. Coin.

**Software:** Son Hoang Nguyen.

**Supervision:** Minh Duc Cao, Lachlan J. M. Coin.

**Validation:** Son Hoang Nguyen.

**Visualization:** Son Hoang Nguyen.

**Writing – original draft:** Son Hoang Nguyen.

**Writing – review & editing:** Minh Duc Cao, Lachlan J. M. Coin.

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
