## [Decision Letter · Decision Letter 0]

9 Jul 2020

Dear Dr Coin,

Thank you very much for submitting your manuscript "Real-time resolution of short-read assembly graph using ONT long reads" for consideration at PLOS Computational Biology.

As with all papers reviewed by the journal, your manuscript was reviewed by members of the editorial board and by several independent reviewers. In light of the reviews (below this email), we would like to invite the resubmission of a significantly-revised version that takes into account the reviewers' comments.

We cannot make any decision about publication until we have seen the revised manuscript and your response to the reviewers' comments. Your revised manuscript is also likely to be sent to reviewers for further evaluation.

Sincerely,

Kin Fai Au

Guest Editor

PLOS Computational Biology

Jian Ma

Deputy Editor

PLOS Computational Biology

Reviewer's Responses to Questions

**Comments to the Authors:**

**Reviewer #1: **The manuscript describes the algorithm, npGraph, offers a way to improve the assembly completeness and visualize the procedure in real-time using Nanopore long reads. On the synthetic and real dataset, the authors show that npGraph yields assemblies of comparative performances to other hybrid scaffolders. It is really a nice tool, which is thought to be popular in sequencing analysis. To make the results stronger in comparison, I give my suggestions and questions.

In Contigs Binning step, by using kmer coverages to determine the contigs membership, is the kmer size sensitive to this?

npGraph is a hybrid scaffolder based on streaming data. In table 1, Unicycler has high accurate assemblies than other algorithms, even it takes long time. Though with unique merits by itself, it would be good to show the performances and benchmarks among hybrid scaffolders with increasing coverage of long reads (Such as 1x -> 5x -> 10x -> 20x).

Another two potentially missing software used in scaffolding would be FastSG (https://github.com/adigenova/fast-sg) and LRScaf (https://github.com/shingocat/lrscaf), which are two differences strategies on hybrid scaffolding. FastSG does the scaffolding step by simulating mate-pair information from long reads, whereas LRScaf is fast hybrid scaffolders on low coverage long reads by using full length information.

As mentioned by authors, npScarf is a greedy scaffolder with only 1 spanning long-read, whereas npGraph requires 3 reads. If improving the required number of long reads for npScarf, would the misassemblies and contiguous of assemblies be decreased? It is reasonable to benchmark the performance between npScarf and npGraph on similar circumstances.

Line 79, “In order to define a customised metric which is sample and fast to calculate”, is this a typo for simple?

In supporting figure S2, the figure is not consistent with the description.

**Reviewer #2: **This paper reports a new streaming pipeline npGraph for hybrid assembly, which uses error-prone long reads to improve fragmented assemblies from short reads. The idea is to use long reads to bridge contigs in the assembly graph from short reads. Tested on simulated and real datasets, the proposed streaming pipeline achieved comparative performances with existing batch-mode hybrid assemblers including Unicycler and SPAdes hybrid, and outperformed the existing streaming approach npScarf. The method is sound and results are convincing. My only doubt about the proposed approach is the significance of having a streaming algorithm for hybrid assembler. There are important real-time analysis of the ONT reads, but I am not so sure about hybrid assembly.

Other specific comments:

1) How the important parameters (the eps and min samples ) for DBSCAN are determined? What's their impact on the performance of npGraph.

2) I assume that the binning results are used to constraint the bride candidates -- bridges are only considered between contigs in the same bin. This needs to be clarified in the methods.

3) The paper mentioned that external binning algorithms including MetaBAT and maxBin can be utilized. Are they already implemented in the pipeline? If not, will that be straightforward to implement?

4) Is it a typo in algorithm 2: set of candidate paths connecting v0 to v2 -> v0 to vn?

5) The relationship between Algorithm 1 & 2 is not well explained.

6) It is unclear how the estimated multiplicity is used for path finding (aren't paths candidates ranked according to the likelihood computed based on long reads to contig alignment?).

7) Line 79, sample and fast -> simple and fast?

**Have all data underlying the figures and results presented in the manuscript been provided?**

Reviewer #1: Yes

Reviewer #2: Yes

PLOS authors have the option to publish the peer review history of their article (what does this mean?). If published, this will include your full peer review and any attached files.

Reviewer #1: **Yes: **Jue Ruan

Reviewer #2: No
---

## [Decision Letter · Decision Letter 1]

21 Oct 2020

Dear Dr. Coin,

Thank you very much for submitting your manuscript "Real-time resolution of short-read assembly graph using ONT

long reads" for consideration at PLOS Computational Biology. As with all papers reviewed by the journal, your manuscript was reviewed by members of the editorial board and by several independent reviewers. The reviewers appreciated the attention to an important topic. Based on the reviews, we are likely to accept this manuscript for publication, providing that you modify the manuscript according to the additional review recommendations.

The revision has addressed all the comments raised by the reviewers. Before we accept your manuscript for publication, I may ask the authors to submit a minor revision to respond a reviewer's suggestion below:

"However, on the table 1 benchmarks, the metrics are measured on batch-mode, I recommended to have a description of how many folds of data to yield these final assemblies. On line 262, there should be a blank between Unicycler and required. On line 305, the unit ‘Kbp’ is no need to italic."

Sincerely,

Kin Fai Au

Guest Editor

PLOS Computational Biology

Jian Ma

Deputy Editor

PLOS Computational Biology

[LINK]

The revision has addressed all the comments raised by the reviewers. Before, we accept your manuscript. I may ask the authors to submit a minor revision to respond a reviewer's suggestion below:

"However, on the table 1 benchmarks, the metrics are measured on batch-mode, I recommended to have a description of how many folds of data to yield these final assemblies. On line 262, there should be a blank between Unicycler and required. On line 305, the unit ‘Kbp’ is no need to italic. I look forward to the publication of this neat piece of software in PLOS Computational Biology after some statement’s corrections."

Reviewer's Responses to Questions

**Comments to the Authors:**

Reviewer #1: I am overall satisfied with the revised manuscript and all the concerns have been answered and resolved. npGraph gives a flexible and reliable opportunity for hybrid scaffolding with streaming data. However, on the table 1 benchmarks, the metrics are measured on batch-mode, I recommended to have a description of how many folds of data to yield these final assemblies.

On line 262, there should be a blank between Unicycler and required. On line 305, the unit ‘Kbp’ is no need to italic. I look forward to the publication of this neat piece of software in PLOS Computational Biology after some statement’s corrections.

Reviewer #2: The authors have addressed all my concerns.

**Have all data underlying the figures and results presented in the manuscript been provided?**

Reviewer #1: None

Reviewer #2: Yes

PLOS authors have the option to publish the peer review history of their article (what does this mean?). If published, this will include your full peer review and any attached files.

Reviewer #1: **Yes: **Jue Ruan

Reviewer #2: **Yes: **Yuzhen Ye
---

## [Editor Report · Decision Letter 2]

30 Nov 2020

Dear Dr Coin,

We are pleased to inform you that your manuscript 'Real-time resolution of short-read assembly graph using ONT long reads' has been provisionally accepted for publication in PLOS Computational Biology.

Best regards,

Kin Fai Au

Guest Editor

PLOS Computational Biology

Jian Ma

Deputy Editor

PLOS Computational Biology

---

## [Editor Report · Acceptance letter]

15 Jan 2021

PCOMPBIOL-D-20-00393R2

Real-time resolution of short-read assembly graph using ONT
long reads

Dear Dr Coin,

I am pleased to inform you that your manuscript has been formally accepted for publication in PLOS Computational Biology. Your manuscript is now with our production department and you will be notified of the publication date in due course.

With kind regards,

Jutka Oroszlan
